# Breeding Milestones Correspond with Changes to Wheat Rhizosphere Biogeochemistry That Affect P Acquisition

Rebecca K. McGrail [1,2,*], David A. Van Sanford [3] and David H. McNear, Jr. [1]

[1] Rhizosphere Science Laboratory, Department of Plant and Soil Sciences, University of Kentucky, 1100 South Limestone, Lexington, KY 40546-0091, USA

[2] Grassland Ecology Laboratory, Department of Plant and Soil Sciences, University of Kentucky, 1100 South Limestone, Lexington, KY 40546-0091, USA

[3] Wheat Breeding Group, Department of Plant and Soil Sciences, University of Kentucky, 1405 Veterans Drive, Lexington, KY 40546-0091, USA

[*] Correspondence: rkmcgrail@gmail.com

**Abstract:** Breeding wheat (*Triticum aestivum* L.) has resulted in small gains in improved nutrient acquisition and use as numerous traits are involved. In this study, we evaluated the impact of breeding on P-acquisition and identified both plant and soil variables that could be used to inform the selection of germplasm with increased P acquisition efficiency. We previously screened a historic panel of winter wheat cultivars for root system architecture and root tip organic acid content when grown in P-deficient solution/agar and used these characteristics together with breeding history to develop a predicted P extraction potential (PEP). We tested the validity of the PEP classification by growing cultivars under sufficient and insufficient soil P conditions. Old, wild-type cultivars had the greatest P utilization efficiency (PUtE) when grown under insufficient P, likely a result of the chemical potential of wild-type (with respect to Rht-B1) cultivars (greater organic acid production) rather than root system size. Wild-type plants had differences in rhizosphere microbial community structure, rhizosphere bicarbonate-extractable P, and bulk soil Fe and Al, indicating the utilization of typically less available P pools. The PEP classification based on the presence of dwarfing allele and era of release offers a path forward for breeding for improved P acquisition.

**Keywords:** breeding; microbial community; nutrient acquisition; phosphorus; rhizosphere; root system architecture; winter wheat

## 1. Introduction

Yield gaps, in which the crop yield potential is not achieved, result from numerous factors including unfavorable weather conditions (temperature and moisture extremes), pests, disease, and nutrient deficiencies. It is estimated that solely improving nutrient balances could close the yield gap in 73% of agricultural regions that report a yield gap exceeding 25% [1]. Improving nutrient balances cannot be accomplished solely through the increased use of fertilizers because fertilizer use efficiencies for macronutrients are low—$\leq 50\%$ for nitrogen, $\leq 10\%$ for phosphorus, and 20–40% for potassium [2]. Estimates suggest that plant available P is limited in 67% of agricultural soils [3]. Therefore, improving the P balances is a major target for closing the yield gap. The reliance on inorganic P fertilizer alone to improve P balance and close yield gaps is volatile and can be economically unattainable for producers, most notably in developing nations [4–7].

The adaptation of germplasm for low fertility soils and/or reduced input production systems is the best path forward [6]. Two major targets for improving the germplasm for these conditions are root system architecture and biochemical mechanisms of P acquisition. Diffusion is the principal mechanism by which plants acquire P, accounting for 89% of all uptake in corn (*Zea mays* L.) [8]. However, uptake via diffusion creates depletion zones around the roots. Depletion zones reduce uptake efficiencies through a decline in nutrient

flux per unit of root surface with time [9]. Large root systems can overcome this problem by increasing the amount of soil explored, therefore increasing the amount of P-containing soil solution the roots come into contact with. Because P uptake is proportional to the root length density, plants can respond by increasing the overall root number or by producing roots with a smaller diameter or root hairs to increase the root surface area. Such improvements in root system architecture require less carbon investment than primary roots and offer a mechanism through which to obtain P without greatly increasing plant stress [10,11]. Genetic variance in lateral root growth in wheat (*Triticum aestivum* L.) genotypes [12], seminal root traits in barley (*Hordeum vulgare* L.) breeding lines [13], and both root hair presence [14] and length [15] in rye (*Secale cereale* L. cv. Petkus II) support the adaptation of grain cultivars for low fertility conditions, either low natural fertility or reduced inputs to use legacy P.

A second, promising target is the biogeochemical response to P-deficiency. Plants can exude a variety of compounds to increase P availability including enzymes, organic acids, and protons. Differences in organic acid or proton production have been reported between crop species [16] and among genotypes within a species [17–20], which are linked to increases in P availability. The organic acid response to a given nutrient condition will vary by plant. For example, under low P conditions, alfalfa (*Medicago sativa* L.) and oat (*Avena sativa* L.) released citrate [21,22] whereas rape (*Brassica napus* L.) released malate [23]. Some cultivars of wheat have been reported to contain citric acid [24] and oxalic acid [19,25,26] in the root tips under P-deficiency. However, not all plant species respond to P-deficiency with increased organic acids. Hoffland et al. [27] found that hedge mustard (*Sisymbrium* officinale Scop.) does not accumulate or excrete organic acids. Differences in enzyme exudation such as phytase, acid phosphomonoesterase, and acid phosphatase have been observed in plant genotypes [28–30], which correlated with increased tissue P content [31]. Additionally, microbes in the rhizosphere can also play a role in enhancing P acquisition. The most reported of these is the association between the majority of land plants and mycorrhiza fungi, which can greatly increase the root surface area [32] and influence nutrient foraging strategies [33]. Rhizosphere bacteria, supported by the greater availability of C in the rhizosphere, have also been shown to mineralize organic P or solubilize inorganic P from soil minerals [34]. Both plant and microbial processes play a role in enhancing P availability in the rhizosphere. Selection for plant traits that influence microbial community assemblages, rather than breeding for the microbial community itself, is a suitable target for advancing plant nutrition [35].

Breeding to improve plant response to nutrient deficiencies and to increase nutrient acquisition is crucial [36]. However, genotype information is useful for breeding if the expressed phenotypic response that resulted in improved performance is also known [37]. The genetic underpinnings of root phenotypic responses are poorly understood, especially in comparison to our increasing knowledge of the plant genome, transcriptome, proteome, and metabolome, and are a bottleneck to advancing breeding [37]. We set out to evaluate the winter wheat cultivar performance when grown in soils with low P availability based on an a priori classification. Cultivars selected for this study were previously characterized by the authors for root system architecture (RSA) and root tip organic acid (OA) content in response to P-deficiency. Based on prior analysis, semi-dwarf wheat varieties (all Rht-1b semi-dwarfs) were found to contain two times less organic acids in root tips under P stress compared to wild-type cultivars [19]. RSA analysis determined that under P stress, old cultivars released prior to 1970 had larger root systems compared to intermediate (1970–1989) and modern (1990–2002) cultivars [38]. RSA and OA production were used to group the cultivars into groups based on the P extraction potential (PEP). We hypothesized that old wild-type cultivars (PEP class 1) would perform best under low P conditions compared to newer semi-dwarf varieties with smaller root systems (PEP class 4).

## 2. Materials and Methods

### 2.1. Plant Growth Conditions

Winter wheat seeds for use in this study were provided by the United States Department of Agriculture Agricultural Research Service National Genetic Resources Program. Pembroke 2016, developed by the Kentucky Agricultural Experiment Station [39], was used as a check cultivar for growth parameters as it has been well-characterized and grown by the University of Kentucky Wheat Breeding Group in a variety of environments. Seeds of Pembroke 2016 were provided by the University of Kentucky Wheat Breeding Group. Cultivars were previously described by the authors [19,38] and classified for this study as shown in Table 1.

**Table 1.** Wheat cultivars selected for this study with previous organic acid (OA) [19] and root system architecture (RSA) [38] study classifications.

| Cultivar | Year Released | Era | Dwarfing Status | OA Potential Class | RSA Potential Class | PEP Class |
|---|---|---|---|---|---|---|
| Goens | 1808 | old | wild-type | high | large | 1 |
| Red May | 1830 | old | wild-type | high | large | 1 |
| Blackhull | 1917 | old | wild-type | high | large | 1 |
| Norin-10 | 1935 | old | semi-dwarf | low | large | 2 |
| Nicoma | 1971 | modern | wild-type | high | small | 3 |
| Baca | 1973 | modern | wild-type | high | small | 3 |
| Prowers | 1997 | modern | wild-type | high | small | 3 |
| Lindon | 1975 | modern | semi-dwarf | low | small | 4 |
| Akron | 1994 | modern | semi-dwarf | low | small | 4 |
| Ankor | 2002 | modern | semi-dwarf | low | small | 4 |

Crider silt loam (fine-silty, mixed, active, mesic Typic Paleudalf; Table S1) was selected for use in this study as it is a highly productive soil with insufficient P (6 mg kg$^{-1}$) underlying 202,343 hectares of land in Kentucky, many of which are classified as prime farmland [40]. Soil was passed through a sieve (<2 mm) and limed to a target pH of 6.4 according to the University of Kentucky's recommendations [41] with reagent grade calcium hydroxide (Fisher Scientific, Waltham, MA, USA). Seeds were germinated by planting one seed per cell of a six-cell plant pack filled with limed soil and placing them into plant flats where they were watered continuously with deionized (DI) water from the bottom. After germination, the flats were transferred to a cold room for six weeks for vernalization.

Following vernalization, the entire contents of each individual cell pack was transferred to the center of a labeled pot (950 cm$^3$ volume; [42]) filled with 800 g of limed soil (target pH = 6.4). Pots (n = 5 per cultivar per condition) were randomly arranged on a greenhouse bench and immediately watered with 100 mL of DI water after transplanting. Experimental conditions (sufficient/insufficient P) were established through the addition of 100 mL of nutrient solution. Nutrient solution for both treatments contained reagent grade ammonium nitrate and potassium sulfate (Fisher Scientific, Waltham, MA, USA). Nutrient solution for plants in the sufficient P treatment also contained reagent grade phosphorus(V) oxide (Fisher Scientific, Waltham, MA, USA). The insufficient P treatment received no P fertilization. Nutrient solutions were prepared in ultrapure DI, and nutrient sources were added according to the rate specified by the University of Kentucky's recommendations [41] scaled to the amount of soil per pot (0.58 g $NH_4NO_3$, 0.03 g $K_2SO_4$, and 0 g or 0.06 g $P_2O_5$). Plants were watered every other day with 100 mL of DI water following the nutrient addition. The volume of water added each day was 50% of the soil's water holding capacity. Granular Marathon (1% imidacloprid) was added to each pot according to the label 22 days post-transplant to treat for aphids.

Plants were harvested 47 days after transplant with a target growth stage of heading (Feekes 10.1). At this stage, approximately 80% of the total P uptake had occurred [43–45]. Plant height was measured from the soil surface to the top of the head on the primary

tiller (excluding awns), and the number of tillers and heads were counted. Biomass was harvested by cutting the stem at the soil surface with a razor blade. Shoot and root biomass was measured after drying at 60 °C.

## 2.2. Soil Collection and Analyses

Plants were depotted by applying slight pressure to the bottom of the inverted pot. An effort was made to segregate and sample the root-associated soil and bulk soil not impacted by roots. The soil that freely broke away from the root system in the depotting process was considered bulk soil, while the soil that was retained by the root system was classified as rhizosphere soil. The roots were collected and placed into a separate labeled paper bag. The rhizosphere soil was sieved (<2 mm) to separate the soil from roots, with roots passing through the sieve collected and added to the root biomass. There was not a large amount of rhizosphere soil, so the decision was made to collect a ~7 g sample and ~50 g sample of rhizosphere soil for microbial phospholipid fatty acid analysis (PLFA) and Hedley sequential fractionation, respectively. The small amount of rhizosphere soil remaining was mixed with the sieved bulk soil and a sample was sent to the University of Kentucky Regulatory Services for analysis (pH, Mehlich-III extractable nutrients, %N, and %C).

Samples for PLFA analysis were stored at −20 °C until lyophilization and thereafter at −80 °C until extraction. Fatty acid methyl esters (FAMEs) were extracted from the rhizosphere soil following the high-throughput methods described by Buyer and Sasser [46]. FAME concentrations were determined using an Agilent 7890 GC (Agilent Technologies, Wilmington, DE, USA) equipped with an auto sampler and flame ionization detector controlled by MIS Sherlock® (MIDI, Inc., Newark, DE, USA) and the Agilent ChemStation software. FAMEs were separated on an Agilent 7693 Ultra 2 column (25 m long × 0.2 mm internal diameter × 0.33 μm film thickness) with a column split ratio of 30:1 using ultra-high-purity hydrogen gas at a flow rate of 1.2 mL min$^{-1}$. The oven temperature was 190 °C, ramping to 285 °C for 10 min and then 310 °C for 2 min at a rate of 10 °C min$^{-1}$ and 60 °C min$^{-1}$, respectively. Blanks (hexane only) were analyzed every ten samples and a 37-component fatty acid mixture (Supelco, Sigma Aldrich, Saint Louis, MO, USA) was used to verify the calibration. FAMEs were identified and their concentrations and percentages calculated using the peak naming table in the Sherlock® microbial identification system (MIDI, Newark, DE, USA).

The rhizosphere soil samples were sequentially extracted for inorganic P following the Hedley procedure as described by Tiessen et al. [47] with slight modification. A 4 cm × 10 cm anion exchange resin (CTL Scientific Supply Company, Deer Park, NY, USA) activated with 0.5 N nitric acid ($HNO_3$) for a minimum of 12 h was used rather than Dowex and exchange resin. No other modifications were made. A flatbed orbital shaker (Barnstead International, Dubuque, Iowa) set at 110 rpm was used to shake the samples, and samples were centrifuged with an Eppendorf 5810R centrifuge (Eppendorf, Hamburg, Germany) at 3800 rpm for 50 min. Inorganic orthophosphate was quantified by ICP-OES (5110, Agilent Technology, Santa Clara, CA, USA), and the concentration was determined from standards of a known concentration. In the Hedley procedure, resin-extractable and bicarbonate-extractable P are plant-available P. The resin-extractable P is directly taken up by plants and the bicarbonate-extractable pool replenishes the resin-available pool.

## 2.3. Plant Digestion and Elemental Quantification

Oven-dried shoot and roots with masses ≥ 0.20 g were ground in a ball mill (Cianflone Scientific Instruments Corporation, Pittsburgh, PA, USA) prior to digestion. Root and shoot samples with masses ≤ 0.20 g were not ground, and the entire sample was digested whole. Grinding such small quantities resulted in minimal material recovery. Tissue samples were digested in open top 50 mL metal-free conical vials using 10 mL of trace metal grade 31.4 N $HNO_3$ in a 1600 W MARS Xpress microwave (Matthews, NC, USA) digestion system at 100 °C for 20 min. Digests were diluted to a final volume of 50 mL with ultrapure DI water

prior to analysis for Al, Ca, Fe, K, Mg, Mn, and P concentrations on an ICP-OES (5110, Agilent Technology, Santa Clara, CA, USA). The instrument was calibrated with a mixed standard and verified every twenty samples along with a calibration blank. An internal standard of yttrium was continuously analyzed to test for instrumental drift and matrix effects. Five samples of varying masses of apple leaf standard reference material (NIST SRM 1515, Millipore Sigma, Burlington, MA, USA) were digested and analyzed to calculate the percent recovery.

Two measures of P uptake and efficiency were calculated from the shoot and root P concentrations. Phosphorus utilization efficiency (PUtE) and root P acquisition efficiency (RPAE) were computed following the equations in Pan et al. [48].

$$PUtE = [shoot\ dry\ mass + root\ dry\ mass\ (g)]/[shoot\ P\ content + root\ P\ content\ (g)]$$

$$RPAE = [shoot\ P\ content + root\ P\ content\ (mg)]/root\ dry\ mass\ (g)$$

*2.4. Statistical Analysis*

Univariate outlier analysis was performed by experimental condition (sufficient/insufficient P) using JMP Pro 16 (Statistical Analysis Systems Institute, Cary, NC, USA) by which any value that was three times the interquartile range beyond the lower and upper quartiles was classified as an outlier. In the sufficient P portion of the experiment, a single Baca (modern wild-type) rep was identified as an outlier for root Ca, root Al, root Mg, and root P, likely due to soil contamination in the digestate. These values were excluded from any further analysis. No measurements were identified as an outlier in the insufficient P portion of the experiment.

Statistical analysis of plant growth metrics, plant nutrients, soil chemical data, measures of nutrient use efficiency, and the total concentrations of microbial biomarker groups (Gram-positive bacteria (G+), Gram-negative bacteria (G−), arbuscular mycorrhizal fungi (AMF), general fungi, actinobacteria, protists) conducted in SAS 9.4 (Statistical Analysis Systems Institute, Cary, NC, USA). PROC GLIMMIX (generalized linear mixed model), which does not require the assumption of normality, was used to test the effects of dwarfing status and cultivar age classification. The experimental condition (sufficient/insufficient P) was statistically significant for 38 of the 60 variables, so all analyses were performed by the experimental condition.

A two-way ANOVA was used to assess the impact of the fixed effects of cultivar dwarfing status and cultivar age with replicate as a random effect. No interaction was specified in the model as the interaction term was the PEP classification. Means were separated using a post-hoc Tukey's honest significant difference (HSD) test ($\alpha = 0.05$). A one-way ANOVA was used to evaluate the PEP class on the measures of efficiency (PUtE and RPAE) with the replicate specified as a random effect. Means were separated using a post-hoc Tukey's HSD test ($\alpha = 0.05$). A complete ANOVA table for all variables is available in Table S2.

The microbial community structure was analyzed in PC-ORD (version, 6.08, MjM Software, Gleneden Beach, OR, USA) using Hellinger transformed [49] microbial biomarker group concentrations in autopilot mode using Sorensen (Bray-Curtis) distances and slow and thorough settings. A multi-response permutation procedure (MRPP) was used with a relative Sorenson distance measure of the PLFA matrix to evaluate whether the cultivar dwarfing status (wild-type or semi-dwarf), cultivar age (old or modern), or PEP classification affected the microbial community structure. The p-values were corrected for multiple comparisons using the Benjamini and Hochberg [50] approach with a false discovery rate of 0.10.

## 3. Results

*3.1. Measures of Phosphorus Efficiency*

Under sufficient soil P, PUtE and RPAE did not differ significantly with the PEP class (Table 2). With insufficient P, PUtE differed significantly with the PEP class. PUtE was

greatest for PEP class 1, composed of the old wild-type cultivars (Figure 1), and was lowest for PEP class 4, which was the modern semi-dwarf cultivars. RPAE did not differ with the PEP class. Because the PEP class is a categorical combination of two factors (plant era and dwarfing status), explanatory factors that inform PUtE were evaluated by both plant era and dwarfing status to ascertain which plant root responses corresponded with either changes in RSA (plant era) or OA production (dwarfing status).

**Table 2.** Average (least squares means ± one standard error) P utilization efficiency (PUtE) and root P acquisition efficiency (RPAE) by the PEP classification under sufficient or insufficient soil P. Parameters not sharing the same uppercase (sufficient P) or lowercase (insufficient P) letter were significantly different (Tukey's HSD, $p < 0.05$).

| PEP Class | Sufficient P | | Insufficient P | |
|---|---|---|---|---|
| | PUtE | RPAE | PUtE | RPAE |
| Class 1 | 897 ± 136 A | 9 ± 13 A | 259 ± 23 a | 41 ± 14 a |
| Class 2 | 412 ± 235 A | 44 ± 19 A | 148 ± 41 ab | 24 ± 25 a |
| Class 3 | 818 ± 140 A | 13 ± 13 A | 204 ± 24 ab | 29 ± 15 a |
| Class 4 | 539 ± 146 A | 23 ± 13 A | 147 ± 23 b | 68 ± 14 a |

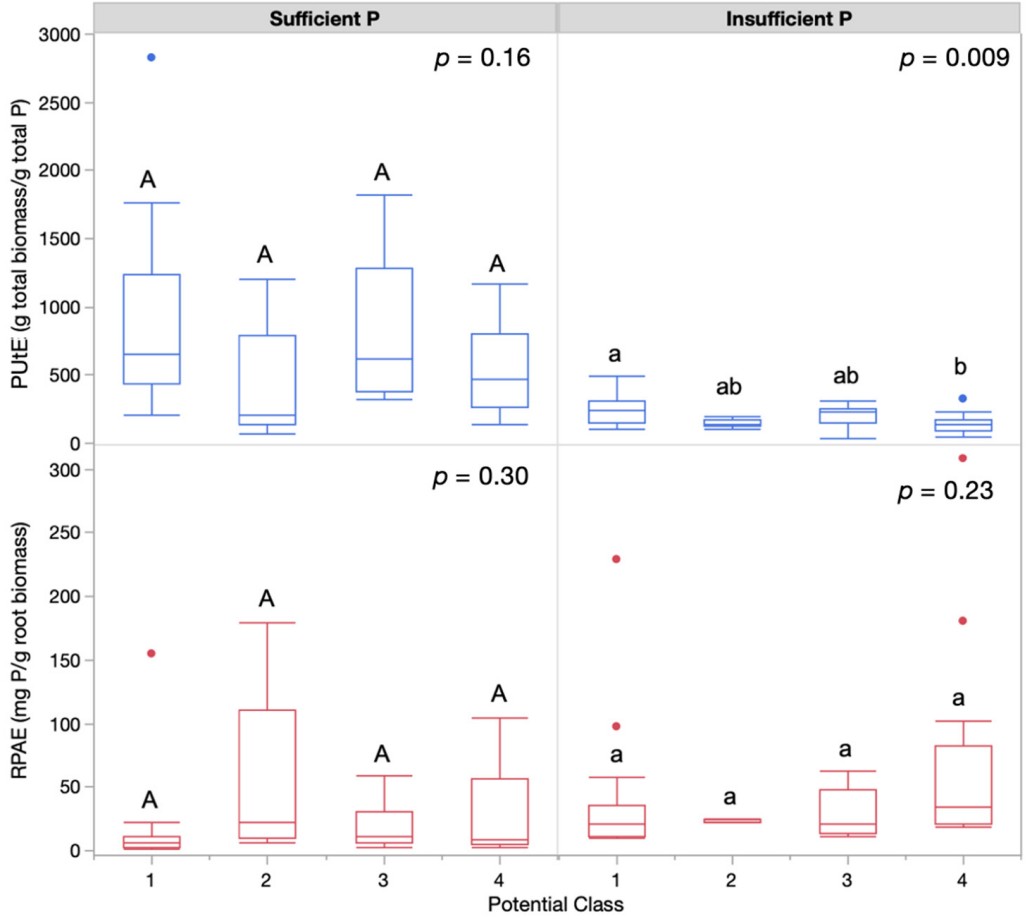

**Figure 1.** Phosphorus utilization efficiency (PUtE; g total biomass g$^{-1}$ total P) and root P acquisition efficiency (RPAE; mg total P g$^{-1}$ root biomass) between the hypothesized PEP classes (n = 15, 5, 15, 15, respectively, per condition) grown with sufficient or insufficient P. Boxes for each parameter not sharing the same uppercase (sufficient P) or lowercase (insufficient P) letter were significantly different (Tukey's HSD, $p < 0.05$).

### 3.2. Plant Growth and Nutrient Content

With sufficient P, cultivar dwarfing status had a significant ($p < 0.05$) effect on the main stem height, shoot mass, and shoot P, root Al, and root Fe concentrations (Table 3). Wild-type cultivars were taller and had more shoot biomass compared to the semi-dwarf cultivars. Wild-types had 29% less shoot P concentration compared to the semi-dwarfs (Table S3). When accounting for the total nutrient content (shoot or root concentration multiplied by shoot or root mass), shoot P did not vary by cultivar dwarfing status (Tables 3 and S5). Root Al and root Fe concentration decreased by 25% and 33% (Table S3) in the wild-type cultivars compared to the semi-dwarf cultivars, however, this did not result in any difference in the total nutrient uptake (Table 3). The total nutrient content differed significantly ($p < 0.05$) in the wild-types compared to the semi-dwarfs for shoot K, shoot Ca, shoot Mg, shoot Mn, shoot Al, and shoot Fe (Table 3). The total nutrient uptake increased in the wild-types by 42% for shoot K, 49% for shoot Ca, 36% for shoot Mg, 36% for shoot Mn, 40% for shoot Fe, and 33% for shoot Al (Table S5). Shoot Mg and root Fe concentrations differed with cultivar age class (Table 3). Old cultivars contained 23% more shoot Mg and 50% more root Fe compared to modern cultivars (Table S4). Again, this difference in concentration did not correspond with a difference in the total nutrient uptake for either shoot Mg or root Fe (Table S6).

With insufficient P, the growth stage, main stem height, and shoot K, shoot Mg, shoot P concentration varied with the cultivar dwarfing status ($p < 0.05$; Table 3). The wild-type cultivars were taller and contained 11% greater concentration (mg g$^{-1}$ tissue) of shoot K compared to the semi-dwarf cultivars (Table S3). P and Mg concentration in the shoots of the semi-dwarf cultivars was 32% and 11% greater, respectively, compared to the wild-type cultivars (Table S3). However, the total nutrient uptake (mg) did not differ between the wild-types and semi-dwarfs for shoot K, shoot P, or shoot Mg (Table S5). Growth stage, number of heads, shoot K, and shoot Mg differed with the cultivar age class (Table 3). Modern cultivars had an average of one head per plant whereas old cultivars had an average of 0.8 heads per plant, which was attributed to a difference in growth stages. Five wild-type plants were not at the targeted stage (three Red May and one Blackhull plant at flag leaf stage and one Red May plant in boot) and therefore did not have any emerged heads. Shoot K and shoot Mg concentration was 11% and 10% greater, respectively in the wild-type cultivars compared to the semi-dwarf cultivars (Table S3). These differences did not result in a differing total nutrient uptake. The total uptake of shoot Fe and root K differed between the old and modern cultivars (Table 3). The total shoot Fe content was 36% greater and the total root K content was 32% greater in the old cultivars compared to the modern cultivars (Table S6).

### 3.3. Bulk and Rhizosphere Soil Chemical Variables

With sufficient P, the bulk soil P and K ($p < 0.05$) differed significantly by cultivar dwarfing status (Table 3). Bulk soil Mehlich-III P was 36% greater for the wild-types compared to the semi-dwarf cultivars (Table S7). Bulk soil Mehlich-III K was 17% less for the wild-types compared to the semi-dwarf cultivars (Table S7).

Under insufficient P, the bulk soil Mg and rhizosphere bicarbonate-extractable P varied by cultivar dwarfing status (Table 3). Rhizosphere bicarbonate extractable P was 16% less in the soil from the wild-types (Figure 2, Table S7). The rhizosphere resin-extractable P varied with cultivar age class (Table 3). Rhizosphere soil from the wild-type plants contained 25% less resin-available P compared to the soil from semi-dwarf plants (Figure 2, Table S7). Bulk soil Mehlich-III Mg was 4% greater for the wild-types (Tables 4 and S7).

**Table 3.** Two-way ANOVA results for the significant explanatory parameters of P utilization efficiency (PUtE) and root P acquisition efficiency (RPAE) by dwarfing status and era of release under sufficient or insufficient soil P. A complete listing of ANOVA results for all variables can be found in Table S2.

| | Sufficient P | | | | Insufficient P | | | |
| --- | --- | --- | --- | --- | --- | --- | --- | --- |
| | Dwarfing Status | | Era | | Dwarfing Status | | Era | |
| | *F*-Value | *p*-Value | *F*-Value | *p*-Value | *F*-Value | *p*-Value | *F*-Value | *p*-Value |
| Plant Growth Metrics | | | | | | | | |
| Growth Stage | 0.53 | 0.4702 | 3.01 | 0.0892 | 4.29 | 0.0440 | 11.8 | 0.0017 |
| Main Stem Height | 12.53 | 0.0009 | 0.50 | 0.4810 | 8.85 | 0.0047 | 1.77 | 0.1902 |
| Number of Heads | 0.06 | 0.8085 | 3.72 | 0.0601 | 1.49 | 0.2286 | 5.86 | 0.0196 |
| Shoot Mass | 5.70 | 0.0212 | 0.20 | 0.6562 | 1.15 | 0.2894 | 0.96 | 0.3327 |
| Nutrient Concentration (mg kg$^{-1}$) | | | | | | | | |
| Shoot K | 0.15 | 0.6970 | 2.58 | 0.1151 | 4.59 | 0.0374 | 5.52 | 0.0231 |
| Shoot Mg | 0.96 | 0.3322 | 12.82 | 0.0008 | 5.76 | 0.0205 | 6.41 | 0.0148 |
| Shoot P | 11.27 | 0.0016 | 0.01 | 0.9275 | 11.62 | 0.0014 | 2.58 | 0.1147 |
| Root Al | 4.87 | 0.0328 | 2.93 | 0.0940 | 0.65 | 0.4228 | 0.09 | 0.7675 |
| Root Fe | 5.57 | 0.0229 | 4.75 | 0.0349 | 0.54 | 0.4663 | 0.00 | 0.9905 |
| Total Nutrient Content (mg) | | | | | | | | |
| Shoot Ca | 7.39 | 0.0092 | 0.00 | 0.9795 | 2.76 | 0.1037 | 1.42 | 0.2402 |
| Shoot K | 5.85 | 0.0195 | 0.09 | 0.7674 | 2.77 | 0.1029 | 3.26 | 0.0776 |
| Shoot Al | 4.60 | 0.0373 | 0.45 | 0.5037 | 3.28 | 0.0767 | 0.67 | 0.4156 |
| Shoot Fe | 7.67 | 0.0081 | 0.61 | 0.4398 | 3.09 | 0.0853 | 5.29 | 0.0261 |

| | Sufficient P | | | | Insufficient P | | | |
| --- | --- | --- | --- | --- | --- | --- | --- | --- |
| | Dwarfing Status | | Era | | Dwarfing Status | | Era | |
| | *F*-value | *p*-value | *F*-value | *p*-value | *F*-value | *p*-value | *F*-value | *p*-value |
| Shoot Mg | 4.41 | 0.0413 | 0.67 | 0.4165 | 0.30 | 0.5847 | 3.06 | 0.0867 |
| Shoot Mn | 4.62 | 0.0370 | 0.03 | 0.8585 | 0.92 | 0.3417 | 2.27 | 0.1391 |
| Root K | 0.81 | 0.3718 | 2.55 | 0.1176 | 0.09 | 7.598 | 4.34 | 0.0428 |
| Bulk Soil Physicochemical Parameters | | | | | | | | |
| P | 5.17 | 0.0278 | 0.01 | 0.9223 | 1.45 | 0.2344 | 1.86 | 0.1798 |
| K | 7.64 | 0.0082 | 0.96 | 0.3317 | 1.06 | 0.3087 | 1.91 | 0.1734 |
| Mg | 0.10 | 0.7535 | 0.91 | 0.3442 | 4.21 | 0.0460 | 0.06 | 0.8071 |
| Mn | 1.19 | 0.2817 | 10.83 | 0.0019 | 0.06 | 0.8070 | 2.93 | 0.0939 |
| Rhizosphere Soil Inorganic P Fractions | | | | | | | | |
| Resin P | 0.13 | 0.7251 | 0.02 | 0.8798 | 0.51 | 0.4798 | 5.42 | 0.0243 |
| Bicarbonate P | 0.44 | 0.5093 | 0.15 | 0.7009 | 6.48 | 0.0143 | 0.04 | 0.8452 |
| Rhizosphere Soil Microbial Biomass | | | | | | | | |
| G+ Bacteria | 10.90 | 0.0019 | 0.13 | 0.7222 | 5.05 | 0.0295 | 0.06 | 0.8035 |
| G- Bacteria | 0.08 | 0.7735 | 7.09 | 0.0106 | 0.09 | 0.7892 | 2.48 | 0.1218 |
| Actinobacteria | 2.76 | 0.1035 | 6.51 | 0.0141 | 0.19 | 0.6634 | 3.43 | 0.0706 |
| Protists | 5.43 | 0.0242 | 1.63 | 0.2083 | 6.96 | 0.0113 | 0.09 | 0.7694 |

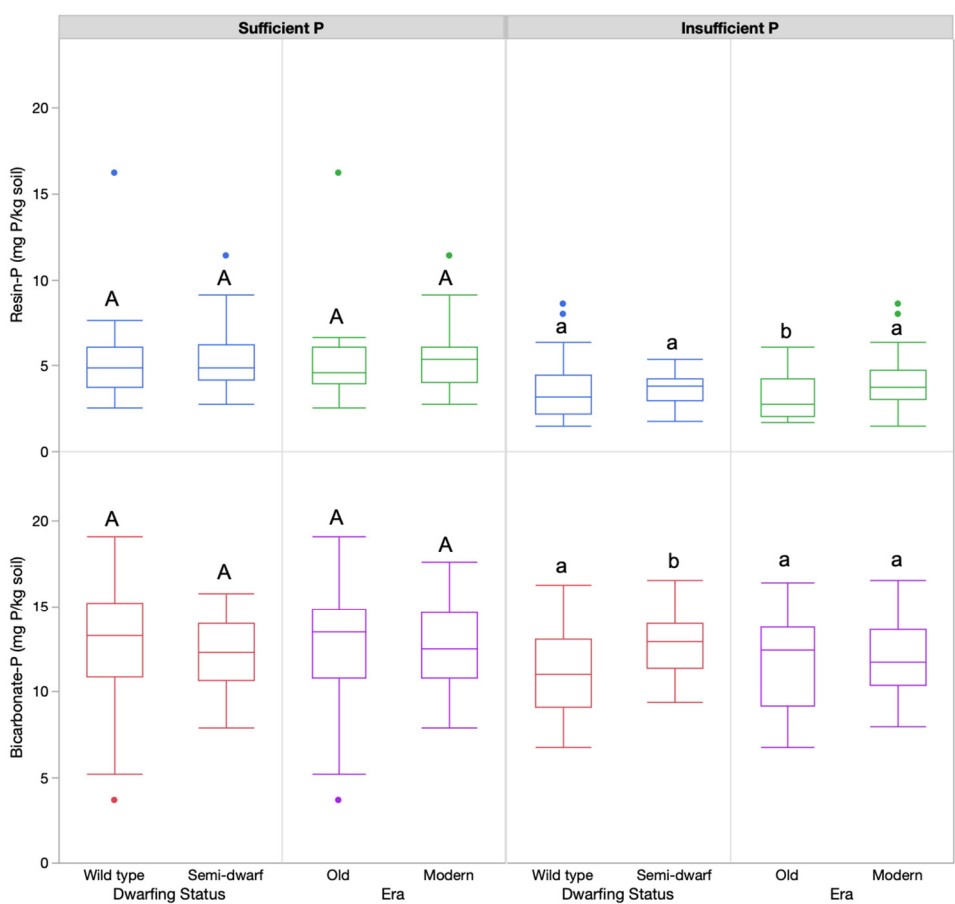

**Figure 2.** Resin-available rhizosphere P (mg kg$^{-1}$ soil) and bicarbonate-extractable rhizosphere P (mg kg$^{-1}$ soil) between the wild-type (n = 30 per condition) and semi-dwarf wheat cultivars (n = 20 per condition) and between the old (n = 30 per condition) and modern (n = 20 per condition) cultivars grown with sufficient or insufficient P. Boxes for each parameter not sharing the same uppercase (sufficient P) or lowercase (insufficient P) letter were significantly different (Tukey's HSD, $p < 0.05$).

**Table 4.** Average (least squares means ± one standard error) Mehlich-III extractable oxides associated with P-sorption by dwarfing status and cultivar era from soil with sufficient or insufficient P. Parameters not sharing the same uppercase (sufficient P) or lowercase (insufficient P) letter were significantly different (Tukey's HSD, $p < 0.05$).

| | **Sufficient P** | | **Insufficient P** | |
|---|---|---|---|---|
| | **Wild-Type** | **Semi-Dwarf** | **Wild-Type** | **Semi-Dwarf** |
| Mg (mg kg$^{-1}$) | 117 ± 2 A | 118 ± 3 A | 119 ± 1 a | 114 ± 2 b |
| Fe (mg kg$^{-1}$) | 84 ± 5 A | 77 ± 6 A | 103 ± 6 a | 97 ± 7 a |
| Al (mg kg$^{-1}$) | 716 ± 9 A | 723 ± 12 A | 762 ± 9 a | 743 ± 12 a |
| | **Old** | **Modern** | **Old** | **Modern** |
| Mg (mg kg$^{-1}$) | 119 ± 3 A | 115 ± 2 A | 117 ± 2 a | 116 ± 1 a |
| Fe (mg kg$^{-1}$) | 75 ± 6 A | 87 ± 5 A | 95 ± 7 a | 104 ± 6 a |
| Al (mg kg$^{-1}$) | 714 ± 12 A | 725 ± 9 A | 741 ± 12 a | 764 ± 9 a |

*3.4. Rhizosphere Microbial Community Structure*

Dwarfing status, era, and PEP class had a significant influence on the microbial community structure in the insufficient P (−P) treatment. The NMDS resulted in a 3-dimensional solution with a final stress of 8.88 after 71 iterations. In the orientation shown (Figure 3), axes 1 and 2 explained 27% and 57% of the variation, respectively, with

an additional 10% for axis 3 (not shown) for a total of 94%. PEP classes 2 and 4 were significantly different from each other and all other PEP classes. Microbial communities in the rhizosphere of the semi-dwarf and wild-type cultivars separated along axis 2 (Figure S1) and were significantly different (MRPP $A = 0.1383$, $p < 0.001$). The rhizosphere of the wild-type plants were strongly correlated with G+ bacteria (axis 2 $r^2 = 0.946$). The effect of era on the rhizosphere microbial community in the insufficient P treatment was significant, but less pronounced ($A = 0.06$, $p = 0.004$; Figure S1), separating along axis 1. The rhizosphere of the modern cultivars was correlated with general fungi (axis 1 $r^2 = 0.905$).

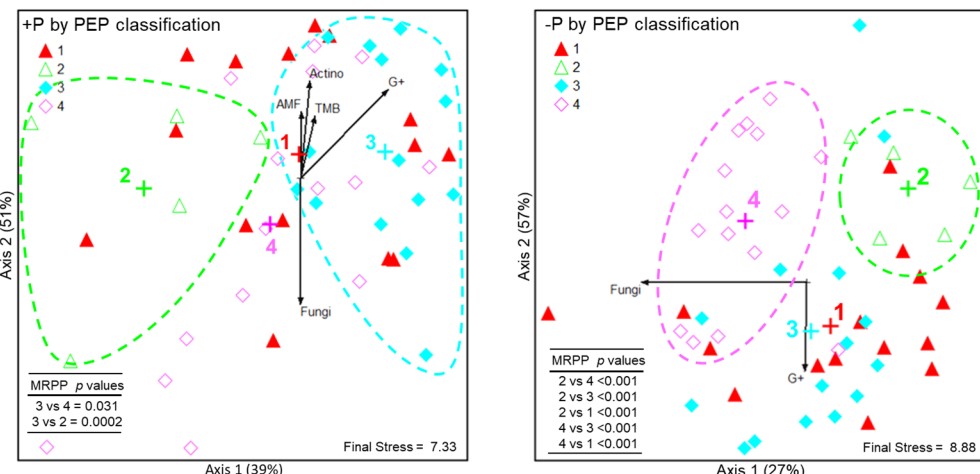

**Figure 3.** Nonmetric multidimensional scaling (NMDS) ordination biplots of Hellinger transformed microbial biomarker group FAMEs showing the separation in microbial communities based on the hypothesized PEP classification in sufficient (+P) and insufficient (−P) conditions. The NMDS resulted in a 3D solution with only the two dominant axes being shown, which cumulatively explained 90% (+P) and 84% (−P) of the overall variance in the dataset. Ellipses show the significant separation in microbial communities based on results from the multiple regression permutation procedure (MRPP). PLFA biomarker group concentrations with $r^2 > 0.30$ between the variable and axis score are displayed as vectors. Vectors indicate strength (length of arrow) and direction of graphed relationships.

Analysis by PEP class in the sufficient P (+P) treatment revealed significant differences in the rhizosphere microbial community structure. The NMDS resulted in a 3-dimensional solution with a final stress of 7.731 after 76 iterations. In the orientation shown (Figure 3), axes 1 and 2 explained 39% and 51% of the variation, respectively, with an additional 7% for axis 3 (not shown) for a total of 96%. The rhizosphere community under PEP classes 3 and 2 separated along axis 1 and were found to be significantly different from each other ($A = 0.158$, $p < 0.001$). PEP classes 3 and 4 were also found to be significantly different, however, the difference was not robust ($A = 0.0469$, $p = 0.031$). There was no effect of era on the rhizosphere microbial community structure in the sufficient P treatment ($A = 0.013$, $p = 0.133$), but the dwarfing status did have a significant, albeit weak, effect ($A = 0.0412$, $p = 0.011$).

Univariate analysis of the microbial biomarker concentrations by P treatment found that with sufficient P, the G+ bacterial biomass was significantly greater in the wild-type compared to semi-dwarf cultivars (12.81 vs. 11.27 nmol g$^{-1}$; Table S7) while the protist biomass was significantly less (0.67 vs. 0.58 nmol g$^{-1}$; Tables 3 and S7). The G− and actinobacteria biomass were significantly greater in the old compared to modern cultivars (19.68 vs. 18.14 nmol g$^{-1}$, and 7.04 vs. 6.50 nmol g$^{-1}$, respectively; Tables 3 and S8). As with the P sufficient conditions, in the insufficient P treatment, the G+ bacterial biomass was significantly greater in the wild-type compared to the semi-dwarf cultivars (12.28 vs. 11.16 nmol g$^{-1}$) while the protist biomass was significantly less (0.63 vs. 0.56 nmol g$^{-1}$; Tables 3 and S7).

## 4. Discussion

In this study, we tested the validity of a predictive P extraction potential (PEP) informed by previous screenings of a historical panel of winter wheat for root system architecture and root organic acid content when grown in P deficient conditions. We hypothesized that plants with larger root systems (cultivars released before 1970) and greater concentrations of organic acids involved in P extraction from soil (wild-type cultivars) would perform better under P insufficient conditions compared to plants with smaller root systems (cultivars released after 1970) and lower concentrations of organic acids involved in P extraction (semi-dwarf cultivars). We found that under P insufficient conditions, the P utilization efficiency (PUtE) was explained by the PEP classes. Old wild-type cultivars (class 1) had the greatest PUtE compared to the modern semi-dwarfs (class 4), which had smaller root systems and lower root tip organic acid concentrations. Our findings agree with those of Pan et al. (2008) who found that soybean root architecture was correlated with PUtE. Our study for the first time, to our knowledge, combines root physical and chemical potentials to predict PUtE in wheat.

There is some debate over the influence of microbial and chemical responses in the uptake and use efficiencies. In a CIMMYT evaluation of 42 semi-dwarf wheat lines, Manske et al. [51] concluded chemical and microbial responses were of less importance for P-uptake efficiency than agronomic traits alone, the number of spikes, and aboveground biomass. However, others have reported that the PUtE genotypic variance could not be explained by agronomic traits alone [52,53]. For example, Hetrick et al. [54] found mycorrhizal colonization of the roots varied with wheat evolutionary history where older cultivars had a greater reliance on the symbiosis than the modern wheat cultivars and even ancient wheat ancestors. The presence of P-solubilizing bacteria in rhizosphere soil has been linked to other factors such as the overall size of the microbial community and organic matter rather than soil P-status [55]. Root exudates such as organic acids can increase the overall microbial community size, which changes microbial P and can influence the transformation of P from more recalcitrant forms [56]. Therefore, the dynamic interaction between plant roots and the rhizosphere microbial community can influence P availability, uptake, and use.

We observed shifts in the microbial community composition that correlate with PUtE under insufficient P, suggesting differing roles of microbial communities in P cycling of the wheat classifications. Under insufficient P, PEP class 4 was distinct from all the other classes (Figure 3), and an abundance of microbial groups differed between the PEP classes. These differences were likely driven by differences in organic acids by the dwarfing status previously found in this panel [19]. Organic acids, more so than sugars, are an important driving factor in shaping rhizosphere microbial communities in forests [57], grasslands [58], and agricultural systems [59]. PEP classes 4 and 2 separated along axis 1 in which PEP class 4 was correlated with a greater concentration of general fungi. Fungal dominance is common with P-stress in which plants do not release organic acids, principally oxalic acid [60]. Classes 2 and 4 were both semi-dwarfs, which produce less oxalic acid under P-stress compared to wild-type cultivars [19]. However, Class 2 was composed only of Norin-10, which had a larger root system compared to the modern semi-dwarfs of Class 4 [38], and consistently performed as an intermediate between the wild-types and the modern semi-dwarfs in organic acid production [19]. Therefore, the fungal dominance of Class 4 is likely to be correlated with the decreased organic acid potential of these modern, semi-dwarf cultivars, and the smaller root system by which it interacted with soil. PEP classes 1 and 3 were similar to each other but separated from classes 4 and 2 along axis 2, which correlated with greater concentrations of G+ bacteria. G+ bacteria are distinct from G− bacteria due to the presence of a peptidoglycan in the cell membrane, which decreases the sensitivity to pH and increases the tolerance of acids within the rhizosphere. The greater concentration of organic acids in the roots of the wild-types (class 1 and 3; [19]) selected for G+ bacteria was capable of tolerating those conditions. NMDS analysis of the community structure was supported by ANOVA analyses on microbial group abundance. Protist abundance in both

sufficient and insufficient P treatments were influenced by dwarfing status in which the semi-dwarf cultivars (regardless of age class, PEP class 2 and 4) had a greater abundance of protists. Protists, which feed on both G+ and G− soil bacteria, have been linked to influencing the production of plant growth promoting compounds by bacteria, particularly indole acetic acid [61]. Additionally, protists are important in soil organic matter mineralization by enhancing nitrogen and P availability and subsequent plant uptake [62–64]. Separation of microbial communities by PEP class, and its correspondence with PUtE, suggests differing mechanisms for increasing P availability.

Cultivars can respond to nutrient availability, or lack thereof, through different mechanisms such as morphological and physiological changes, despite belonging to the same species [65]. In this study, differences in PUtE by PEP class can be attributed primarily to changes in both the root morphology (root system size as a function of old vs. modern cultivars) and chemical potentials (organic acid production in wild-types vs. Rht-loci semi-dwarfs), which affect nutrient acquisition. Mechanisms that resulted in similar shoot P concentrations between PEP class differed by the presence of a dwarfing allele or era as evidenced by the soil physicochemical properties. Under insufficient P conditions, rhizosphere soil bicarbonate-extractable P was lower in soils from the wild-types compared to the semi-dwarfs (Figure 2). Because plant-available resin-P was similar between the wild-types and semi-dwarfs (Figure 2) at the time of harvest, the bicarbonate-extractable P pool replenished the plant available resin-P pool in the rhizosphere of the wild-type cultivars. The decrease in bicarbonate-available P was accompanied by a quantitative increase in Mehlich-III extractable bulk soil Fe and Al concentrations under wild-type cultivars (Table 4). This supports the replenishment theory as bicarbonate-extractable P reflects the P sorbed to soil oxides including Fe, Al, and Mg [66]. Wild-type cultivars can have large or small root systems depending on the era of the cultivar release [38], but universally have high OA potential [19], suggesting that the quantitatively greater bulk soil Fe, Al, and Mg concentrations were due to the destabilization of soil oxides by root-derived organic acids, most likely oxalic acid, and the release of P. While citric and malic acids are usually correlated with P-stress, greater oxalic acid concentrations have been reported in the roots and root exudates of a variety of plants when grown in P-limited soils [17,67]. Oxalic acid requires less carbon and energy to synthesize, and it is not as important a metabolite in plant physiological processes, unlike citric acid [17]. Furthermore, oxalic acid is not as favorable a carbon source for rhizosphere microorganisms compared to citric or malic acids, and therefore could persist in soils longer to chelate metals (e.g., Fe and Al) and liberate P in the process [68,69]. The quantitively increased concentrations of bulk soil Fe, Al, and Mg, coupled with a decrease bicarbonate-extractable P, suggests that wild-type cultivars likely used organic acids, principally oxalic acid, to release soil P.

As older varieties of wheat have been reported to have larger root systems compared to modern varieties, it is further likely that older cultivars would be capable of exploring more soil. Differences in the total shoot and root nutrient concentrations between the old and modern cultivars provide additional support for the increased exploration of soil and utilization of the bicarbonate-extractable P pools including P sorbed to oxides [70] and secondary clay minerals [71,72]. Old cultivars had a greater shoot Fe content and root K content (Table S6). Landraces and old cultivars have been found to have a larger root system than that of modern cultivars [38,73–75]; therefore, the nutrient influx of old cultivars was likely greater. Because all nutrient uptake was not greater in old cultivars, another mechanism other than greater soil exploration can be attributed to the increased uptake of Fe and K. It is more likely that the increased exploration, coupled with differences in the exudation of sugars, enzymes, and organic acids, is responsible for the increased uptake of Fe and K. Through our previous work with organic acid production in this panel, we believe that oxalate is the most likely exudate responsible for the increased Fe and K. As Fe is solubilized by organic acids including oxalate, increased soluble Fe would be present in the soil solution for uptake. Oxalate is an important counterion to Na and K cations [76], and K would remain within the root with the exudation of oxalate. Oxalate production and

release in response to P-deficiency has been found to correspond to increases in soil solution K through the release of K from secondary clay minerals [77–79] such as vermiculate and mica, which were present in the soil we used [80]. Therefore, increased root K and shoot Fe could be indicative of increased oxalate production in the old cultivars.

**5. Conclusions**

Overall, our findings are suggestive that old, wild-type cultivars are capable of extracting more soil P through increased root exploration (trait of old cultivars) and increased exudation (trait of wild-type cultivars), which selects for microbial groups such as fungi and G+ bacteria, capable of enhancing soil P turnover. The unique nature of our panel, covering two major breeding milestones—the introduction of semi-dwarfism and the use of increased inputs (i.e., inorganic fertilizers, pesticides, herbicides, and irrigation use) in germplasm development following the Green Revolution—helps us to better understand variance in P-acquisition mechanisms. Our panel included both wild-types and semi-dwarfs in modern and old age classes, which is a necessary departure from other studies to better understand the mechanisms for P-acquisition to advance breeding efforts. Here, we measured numerous variables—plant, soil chemical, and microbial—that, together with our previous research, improved our understanding of the P-stress response in soil and identified variables that could be measured within breeding programs to identify cultivars capable of using soil P and reducing the amount of inorganic fertilizer necessary for growth. The greater variation in PUtE observed in PEP classes 1 and 3 (wild-types) offers a potential for future breeding work. While the classes represented in this paper were based only on RSA and OA potentials, our prior definition of PEP [81] can be used to expand the number of chemical and physical traits desired or considered important, thereby adding additional classes in the system of choice such as enzymes, seed P content, or nutrient ratios. While our research has provided a system for classifying genotypes on the basis of P acquisition, this is simply a first step in an area that has been neglected by conventional, yield-centric breeding programs. It is likely that as more genotypes are screened, the system will evolve to consider other factors and eventually, proxies such as single nucleotide polymorphisms associated with P-acquisitive phenotypes will be available for breeders to use. We conclude that wild-types, both modern and old, provide a reservoir of traits to improve germplasm for low input conditions and show that phenotyping winter wheat roots in early development stages, as was done in our earlier investigative work, is representative of their performance nearing maturity.

**Supplementary Materials:** The following supporting information can be downloaded at: https://www.mdpi.com/article/10.3390/agronomy13030813/s1, Table S1: Initial physicochemical properties of Crider silt loam soil; Table S2: Complete statistical results of two-way ANOVA for explanatory parameters of P utilization efficiency (PUtE) and root P acquisition efficiency (RPAE) by dwarfing status and era of release under sufficient and insufficient soil P; Table S3: Average (least squares means ± one standard error) growth metrics and tissue nutrient concentration by cultivar dwarfing status; Table S4: Average (least squares means ± one standard error) growth metrics and tissue nutrient concentration by cultivar age class; Table S5: Average total nutrient removals per plant component (least squares means ± one standard error) by cultivar dwarfing status; Table S6: Average total nutrient removals per plant component (least squares means ± one standard error) by cultivar age classification; Table S7: Average (least squares means ± one standard error) soil physicochemical data and microbial group concentration by cultivar dwarfing status; Table S8: Average (least squares means ± one standard error) soil physicochemical data and microbial group concentration by cultivar age classification; Figure S1: Nonmetric multidimensional scaling (NMDS) ordination biplots of Hellinger transformed microbial biomarker group FAMEs showing the separation in microbial communities based on dwarfing status (semi-dwarf vs. wild-type) and era (modern vs. old) in insufficient (−P) or sufficient (+P) conditions.

**Author Contributions:** Conceptualization, R.K.M. and D.H.M.J.; Methodology, R.K.M., D.A.V.S. and D.H.M.J.; Software, D.H.M.J.; Validation, R.K.M. and D.H.M.J.; Formal analysis, R.K.M. and D.H.M.J.; Investigation, R.K.M.; Resources, D.A.V.S. and D.H.M.J.; Data curation, R.K.M.; Writing—original draft preparation, R.K.M.; Writing—review and editing, D.A.V.S. and D.H.M.J.; Visualization, R.K.M. and D.H.M.J.; Supervision, D.H.M.J.; Project administration, D.H.M.J.; Funding acquisition, D.H.M.J. All authors have read and agreed to the published version of the manuscript.

**Funding:** This research was funded by the United States Department of Agriculture National Institute of Food and Agriculture Grant 2016-67019-25281.

**Data Availability Statement:** All generated data for this experiment can be requested by contacting the corresponding author.

**Acknowledgments:** The authors thank Joe Kupper of the University of Kentucky Rhizosphere Science Lab for their assistance in the PLFA sample extraction and analysis and John Connelley of the University of Kentucky Wheat Breeding Group for his assistance in germinating and vernalizing the plants. We appreciate the United States Department of Agriculture Agricultural Research Service National Genetics Resource Program for providing the seed for use in this study.

**Conflicts of Interest:** The authors declare that the research was conducted in the absence of any commercial or financial relationships that could be construed as potential conflicts of interest.

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
