# Peer review of "Breeding Milestones Correspond with Changes to Wheat Rhizosphere Biogeochemistry That Affect P Acquisition"

_agronomy, doi:10.3390/agronomy13030813_

Round 1

Reviewer 1 Report

Dear authors,

thank you for this huge work. It was quite hard to read it - for me, as I am a practical wheat breeder. The results are very intersting - maybe you can improve the conclusions and discussion. Good luck - see my comments enclosed!

I really wonder whether there would be high yielding, tall genotypes. 

I had the idea that testing/selection all breeding generations just im P-deficient soil would be the best and easiest forward strategy to breed for better P aquisition. The challenge is to find the most suitable starting material in order not to loose too much yield. 

As I am a breeder, I firstly think about yield - and harvest Index. The old and modern cultivars extracted the same amount of Phosphorus- but probably the new varieties are higher yielding - anyhow, probably, there exist some traits in tall (only older???) varieties that can be exploited - although the way is not so easy, I admit.

It would be fine to comparably analyse the results of screening experiments on soils with low and sufficient P in the future... so my idea.

Alle the best to your future work!

Reviewer 2 Report

The paper deals with an important and interesting topic of biogeochemical conditions of wheat rhizosphere on P acquisition. I would recommend it for publication after the revision.

Here are some suggestions that could be considered for further improvement of the manuscript.

Abstract

Aside from descriptive findings, the abstract should include results i.e. data of the relevant investigated parameters.

Introduction

Lines 35-37: Beside the data on low fertilizer use efficiency for N, P and K in the topics, do the authors have found some studies that reported on FUE in the other parts of the world? Is the low FUE of the macronutrients solely a problem in developing countries? I should think not.

Lines 39-41: Inorganic P fertilizers are not the only source of phosphorus. If there are reasons that limit the application of organic fertilizers or the other sources of P are less effective, it should be also mentioned, regardless of the development status of a nation. The improvement of P utilization efficiency is of interest not only for farmers in developing countries who cannot afford inorganic fertilizer, but also for organic production and low-input agriculture everywhere, as the authors stated in lines 42-43.

Lines 81-99: This part includes the previous studies on which this study follows up, which is good, but it is read a bit awkward - written in a style that is more appropriate for Materials and Methods. Reference to Table 1 should be definitely moved to MM. I suggest to just briefly report the previous findings here (e. g. under P stress semi-dwarfs contained 2x less OAs in roots than wild-types; root size decreased with the time of the cultivar release; explanation of PEP) and then give reasons for and the aim(s) of the study. The authors could then give more information about the selected varieties and the criteria for their classification in MM.

Since the study touches upon the microbial structure in rhizosphere under different P content, some previous findings related to the role or effect of rhizosphere microbial community on P availability could be included in the introduction.

Materials and Methods

Table 1 I am afraid that only one variety in PEP2 group could bias the results, despite the use of statistics for unbalanced data.

Lines 106-107: It would be better to first mention that Pembroke 2016 was used as a check variety (lines 133-135) and then provide the information about the provision of the seeds. Otherwise, it appears the name of the variety comes out of nowhere, and it is not clear what is its purpose.

Line 116: Please define the abbreviation DI when first mentioned “with deionized (DI) water from the bottom.”

Lines 122-124: “Experimental conditions (sufficient or insufficient P) were established through the addition of 100 mL of nutrient solution containing reagent grade ammonium nitrate and potassium sulfate (Fisher Scientific, Waltham, MA).” Is 100 mL of solution added to both sufficient and insufficient P treatments? Does OR in “(sufficient or insufficient P)” means AND? It’s confusing. Likewise, does OR in line 204 and 218 means AND?

Line 129: Please add “g” after “0”.

Line 131: Please provide the active ingredient for Marathon 1% G.

Line 152: Sample of rhizosphere soil was taken for phospholipid fatty acid analysis (PLFA) but which one, and the results of this analysis were not so obvious. The same remark is for fatty acid methyl esters (FAMEs) lines 156-169. It was indeed said that the PLFA matrix was used for nonmetric multidimensional scaling and showed in Figure 3 and Figure S1, but this should be much better presented and explained.

Table S2 and Table 3: What particular growth stage was measured, compared and how? The plants were harvested when they reached heading stage. Did the authors compare the groups for the number of days required to reach heading stage? It is not clear. Please, provide more information in MM.

Page 10, paragraph 1 (no line numbers): Number of ears per plant could not be compared if one group was at heading stage and the other group of wheat plants was at flag leaf stage when their ears have not emerged. This methodology is just not acceptable. The same is for stem height. At the flag stage, the stem is still growing (elongating), and it continues so until the end of the heading stage. It is not proper to measure and compare stem height at two different growth stages.

Line 143 (the second sequence of line numbers staring from 1): Table S7 should be corrected to Table S6.

Line 143: A verb is missing in “Landraces and old cultivars have been found to a larger root system...”

Lines 186-187: I might have missed something, but how this study correlated phenotyping roots in early development stages (and what exact stages were considered here as early?) with field (this experiment was performed in pots?) performance (which traits or parameters?) near maturity?

Conclusion

Conclusion should be divided from discussion into as a separate section.

Reviewer 3 Report

McGrail and her colleagues evaluated the impact of breeding on P-acquisition and identified both plant and soil variables that could be used to inform selection of germplasm with increased P acquisition efficiency. The conclusions are supported by the data, and the submitted manuscript is written clearly and general interest to the readers. I suggest to accept this manuscript.
